

# A federated learning framework based on transfer learning and knowledge distillation for targeted advertising

Caiyu Su[1], Jinri Wei[1], Yuan Lei[2] and Jiahui Li[3]

[1] Guangxi Vocational & Technical Institute of Industry, Nanning, Guangxi, China
[2] Universiti Pendidikan Sultan Idris, Tanjong Malim, Malaysia
[3] Guangxi University of Foreign Languages, Nanning, Guangxi, China

## ABSTRACT

The rise of targeted advertising has led to frequent privacy data leaks, as advertisers are reluctant to share information to safeguard their interests. This has resulted in isolated data islands and model heterogeneity challenges. To address these issues, we have proposed a C-means clustering algorithm based on maximum average difference to improve the evaluation of the difference in distribution between local and global parameters. Additionally, we have introduced an innovative dynamic selection algorithm that leverages knowledge distillation and weight correction to reduce the impact of model heterogeneity. Our framework was tested on various datasets and its performance was evaluated using accuracy, loss, and AUC (area under the ROC curve) metrics. Results showed that the framework outperformed other models in terms of higher accuracy, lower loss, and better AUC while requiring the same computation time. Our research aims to provide a more reliable, controllable, and secure data sharing framework to enhance the efficiency and accuracy of targeted advertising.

## INTRODUCTION

The emergence of the internet has revolutionized the landscape of advertising, offering an unprecedented speed and reach for delivering promotions to a global audience (*Liu et al., 2019*). As compared to traditional mediums, internet advertising boasts higher interactivity and visual appeal, making it a favored tool among marketers. The advent of smart mobile devices has further facilitated the promotion of internet advertisements, making it more accessible to a wider range of users (*Zu, 2022*). As users navigate the web or social media, they are bombarded with a plethora of advertisements, providing them with diverse information and enhancing their experiences. However, the diverse backgrounds of users present a challenge for indiscriminate advertisement delivery methods, as they often fail to meet the personalized needs of users, and inappropriate advertisements can even cause annoyance (*Altulyan et al., 2019*; *Malviya et al., 2022*; *Namasudra et al., 2017*). To address this issue and maximize advertising economic benefits, targeted advertising recommendation systems have emerged as a solution, providing users with personalized advertising experiences that align with their unique interests and needs.

Corresponding authors
Jinri Wei,
Weijinri_200501@163.com
Yuan Lei, leiyuandata@gmail.com

In recent years, the development of the internet has had a significant impact on the advertising industry, making advertising ubiquitous in people's lives and making online advertising increasingly popular. According to a report on the Chinese advertising market, the scale of the Chinese internet advertising market in 2020 was nearly CNY 500 billion, with a year over year (YoY) growth rate of 13.85%. A report by the *Interactive Advertising Bureau (2022)* states that in 2021, the total revenue for internet advertising in the United States reached an astonishing $189.3 billion, with a YoY growth of 35.4%. However, as the advertising industry continues to evolve, advertisers are gaining access to more and more user information, and traditional centralized machine learning (ML) is struggling to cope with the increasing communication costs and longer transmission delays (*Wu, 2019*; *Yang & Song, 2022*). At the same time, users are beginning to realize that their online behavior is being monitored and that their personal privacy information is being collected and may be shared (*Duan, Ge & Feng, 2022*). As people become increasingly concerned about privacy, some consumers are taking proactive measures to protect their privacy information and prevent merchants from attempting personalized recommendations. The accumulation of these issues has led to the formation of data silos in the advertising industry, making it challenging for advertisers to manage the models and data heterogeneity (*Liu et al., 2021b*).

Federated learning presents a decentralized and secure approach to data sharing in the advertising industry (*Yang et al., 2019*). This method stores local data solely on individual nodes and trains machine learning models locally. The local models' parameters are then aggregated on a central server, with the aim of generating more accurate global models. By transforming the data sharing issue into a local model parameter sharing problem, federated learning effectively resolves data privacy concerns and minimizes the cost of data transmission (*Gholizadeh & Musilek, 2022*). However, in the advertising industry, due to differences in data collection and usage by different advertisers, such as data format, quality, and quantity, the model heterogeneity problem arises, which makes it difficult for advertisers to guarantee data quality. How to fully consider the impact of model heterogeneity while ensuring that various advertisers can effectively participate in the global model training has become a new focus of attention.

Transfer learning can effectively address the challenges of model heterogeneity in the advertising industry (*Bozinovski, 2020*; *Kumari et al., 2021*; *Qu et al., 2022*). Transfer learning is the ability to apply information learned in one environment to new environments. Therefore, this method leverages knowledge obtained from related fields to support the current task, even in the presence of known data sample shortages, data incompleteness or inaccurate information, transfer learning can still help the global ML model learn better from data samples (*Melnikov et al., 2020*). As a result, transfer learning greatly alleviates the problem of model heterogeneity and uneven data quality among advertisers.

## Our contribution

Our motivation behind this work is to overcome the challenges posed by isolated data islands and the heterogeneity of data in the advertising industry. We believe that addressing these issues is crucial to advancing our understanding of personalized

advertising recommendations, while also prioritizing data privacy, user experience, and advertising conversion rates. To achieve these goals, we propose a federated learning framework called Fed-TLKD, which incorporates transfer learning and knowledge distillation. This approach is exciting because it offers a promising solution to complex advertising problems and opens up new avenues for research and innovation in the field.

- In the context of targeted advertising, this article proposes the integration of federated learning to resolve the issue of isolated data islands. Additionally, the combination of federated learning with transfer learning is presented as a solution to tackle challenges posed by model heterogeneity. Specifically, a transfer learning C-mean clustering algorithm is proposed based on maximum mean discrepancy that better evaluates the distribution differences between local ML parameters and global ML parameters, achieving more effective targeted advertising.

- We propose a dynamic selection algorithm that utilizes knowledge distillation and weight correction to ensure effective participation of various advertisers in the global ML training process. Our innovative approach not only addresses the impact of model heterogeneity, but also guarantees fair participation of each client while optimizing weight assignment to enhance the overall performance of the global model. This algorithm represents a significant advancement in the field of federated learning and has the potential to revolutionize the way we approach collaborative machine learning.

- We put the framework to the test with various datasets and evaluated its performance based on accuracy, loss, and AUC metrics. The results of our experiments speak for themselves, as the Fed-TLKD framework outperforms other models with higher accuracy, lower loss values, and better AUC values, all achieved within the same computation time. These findings demonstrate the potential of our framework to significantly enhance the performance of machine learning systems in real-world applications.

The subsequent chapters of this article are organized as follows: "Literature Review" presents a comprehensive review of related literature in the field, "System Model and Problem Description" provides a systematic introduction to the system model and defines the problem, "Fed-TLKD Framework" delves into the intricacies of the Fed-TLKD framework, explaining its mathematical underpinnings in detail, and "Experimental Result" evaluates the performance of the Fed-TLKD framework using two datasets and compares it with other existing federated learning models.

## LITERATURE REVIEW

As the internet develops, the amount of advertising data has rapidly increased. The conventional approach of machine learning has become inadequate in satisfying the demands of the advertising industry. As a result, this has led to high costs of communication and low computational efficiency in the advertising sector. Therefore, many technologies have been introduced to the application scenarios of advertising delivery. In the following section, a systematic and layered literature analysis will be carried

out, encompassing the perspectives of traditional machine learning, federated learning, and transfer learning.

## Machine learning in targeted advertising

In advertising targeting scenarios, machine learning is widely adopted as a method for improving the targeting process (*Liu et al., 2021a*; *D'hooge et al., 2023*; *Lee et al., 2019*; *Fang et al., 2022a*). In advertising targeting scenarios, machine learning typically centrally receives data uploaded from local models, then distributes the results of the training equally to various advertiser nodes. As *Simsek & Karagoz (2019)* further enhance the user portrait through using followee/follower property of microblogs, the experimental results indicated a significant level of accuracy. However, *Simsek & Karagoz (2019)* found that the follower property of microblogs cannot completely represent the user's preferences because the context differences of social media and social networks also produce different requirements. Therefore, they proposed a novel user model that incorporates various aspects of a user's message such as text, title, network links, and topic tags, among others, to generate a comprehensive representation of the user (*Simsek & Karagoz, 2020*). However, *Jeong, Kang & Chung (2021)* proposed that only considering the user's preferences on one platform for advertising recommendations is not enough, and it does not meet the expectations of various users. Therefore, the authors proposed an online advertising recommendation system that leverages the user's consumption history. The system calculates the similarity between users and predicts their preferences for items by comparing the rating history of similar users (*Jeong, Kang & Chung, 2021*). *Zhou et al. (2019)* to prevent misleading advertising from stimulating consumption, they set up a three-party game model to analyze the relationship between the government, advertising platforms, and users to prevent users from making any regretful actions due to the launch of misleading advertising.

In summary, machine learning is indeed an effective method in advertising targeting, but in practical application, there is still a certain accuracy gap. Moreover, traditional machine learning approaches have a limitation where central servers have complete control over both the data content and the training process, which raises concerns over the potential risk of privacy breaches.

## Federated learning in targeted advertising

The integration of federated learning offers robust technical support for the sharing of data among advertisers, enabling them to make informed decisions and improve their targeting strategies (*Zhang et al., 2022a*; *Zhang & Li, 2022*; *Vivona et al., 2022*). Under the support of federated learning, each advertiser node is very flexible and can independently collect data, and the training process does not require the uploading of real data to the central server. Recently, numerous academic researchers have presented a plethora of federated learning frameworks aimed at enhancing the delivery of targeted advertising. For example, *Jiang et al. (2021)* proposed a federated sponsored search auction mechanism based on Myerson's theorem, used for recommending satisfactory ads to users in a short time (*Epasto et al., 2021*). *Epasto et al. (2021)* designed a privacy-enhanced solution by

combining IBA algorithm and federated learning, which can provide privacy guarantee while achieving competitive performance. However, *Liu et al. (2021c)* believe that in considering the protection of privacy data, it is also necessary to pay attention to data exchange among advertiser nodes to ensure the revenue of advertiser nodes. Therefore, the authors proposed an optimized construction method for the targeted advertising delivery scenario, based on a federated learning-enabled gradient boosting decision tree (GBDT) model. The method takes into account data islands and irregular and complex flow data, and introduces a new federated voting mechanism called FedVoting. This mechanism aggregates the ensemble of GBDT models that are protected by differential privacy, through multiple training, cross-validation, and voting processes. As a result, it can generate the optimal model that strikes a balance between performance and privacy protection. The experiments indicate that the mechanism has a high accuracy in long-term prediction. However, relying solely on the collected data for user analysis is often not enough, because user data is flowing, changing, and irregular. Therefore, predicting the original advertising click-through rate (CTR) of users is very necessary. *Wu et al. (2022)* proposed a federated joint original advertising CTR prediction method called FedCTR, which can protect privacy in a way that collects features from cross-platform users.

In the scenario of targeted advertising delivery, the heterogeneous content of data samples and models at different nodes is an important issue. However, most federated learning frameworks overlook this problem, resulting in defects in practical applications. We need to seek more comprehensive and integrated solutions to solve this problem.

## Transfer learning in targeted advertising

Ignoring the impact of heterogeneity, the models we train are often difficult to communicate with. At the same time, there are problems such as data sample missing and poor sample quality in the sample dataset, which further damage the quality of the model. To address this issue, many scholars propose that transfer learning can be used (*Devi, Namasudra & Kadry, 2020*; *Zhang et al., 2022b*; *Fang et al., 2022b*; *Oslund et al., 2022*). For example, *Hong et al. (2019)*, to make advertising recommendations intelligent, understand that videos often come with advertising recommendations, which is an important part of it. Therefore, researchers proposed a model based on transfer learning for intelligent advertising recommendations, and the experimental results indicate that its accuracy is quite impressive (*Jiang et al., 2021*; *Hong et al., 2019*). However, due to the complexity and diversity of advertising data, the researchers found that feature transfer alone cannot uncover the relationships between features in advertising data. Hence, *Jiang et al. (2021)* proposed a click rate method that leverages multi-view feature transfer, and the results of their experiments showed that it outperforms many other advertising click-through rate prediction methods. However, they did not consider the communication time and time cost.

*Lian et al. (2019)* considered that many personalized advertising recommendation studies had the problem that only some labeled products could be recommended during video playback, which meant that it couldn't recommend more products that users really liked to them. Therefore, for the image classification model learned on the large dataset, a

**Table 1 Comparison of different frameworks.**

| Literature | Isolated data island | Model heterogeneity | Calculate time cost |
|---|---|---|---|
| *Simsek & Karagoz (2019)* | × | × | × |
| *Jeong, Kang & Chung (2021)* | × | × | ✓ |
| *Zhou et al. (2019)* | × | × | ✓ |
| *Kumari et al. (2021)* | ✓ | × | × |
| *Zhang et al. (2022a)* | ✓ | × | ✓ |
| *Lian et al. (2019)* | × | ✓ | ✓ |

method is proposed, by transferring the pre-trained deep image classification to solve the scene object recognition problem in the specific task of the model of movies, TV dramas, short videos, and other TV programs, the experiment shows that the model is more effective and efficient (*Lian et al., 2019*). *Manchanda et al. (2019)* combine transfer learning and domain adaptation to use the similarity between users to transfer information from users with enough data to users without any active data, thus reducing time costs.

As shown in Table 1, there are many aspects of research on targeted advertising, but a convincing solution for addressing the problems of isolated data island and model heterogeneity caused by the large amount of data in advertising delivery has not yet been found. Additionally, our use of literature analysis tools also confirms this viewpoint. As shown in Fig. 1, in the Web of Science Core Collection, by using "federated learning", "advertising", and "transfer learning" to extract all literature from January 2018 to December 2022 and selecting keywords that appeared more than 100 times, we used the VOSviewer tool to generate a relationship graph of these keywords. It is clearly visible that the relationship between "advertising" and the other two keywords is weak, proving that few researches connect the three together.

# SYSTEM MODEL AND PROBLEM DESCRIPTION

## System model

As shown in Fig. 2, the framework is composed of two entities, the central server, and the advertiser provider. The communication between these two entities only transfers local ML model parameters, while the actual data is only stored locally in the advertiser provider. The functions of these two entities are as follows:

(1) Central server is the core processing node for personalized advertising delivery and does not receive the original data collected by the advertiser, only receiving local ML model parameters. In our framework, it is responsible for the most difficult personalized federated learning calculations, such as transfer learning, knowledge distillation, and the calculation of the optimal aggregation model, *etc*. We assume that it has infinite computational power and storage space, in other words, it can support any calculation.

(2) Advertiser providers are responsible for collecting a large amount of raw data during the customization process of crowd-targeted delivery strategies, and only storing it in local storage units to form a local dataset $D_{local}$. At the same time, we define the public dataset
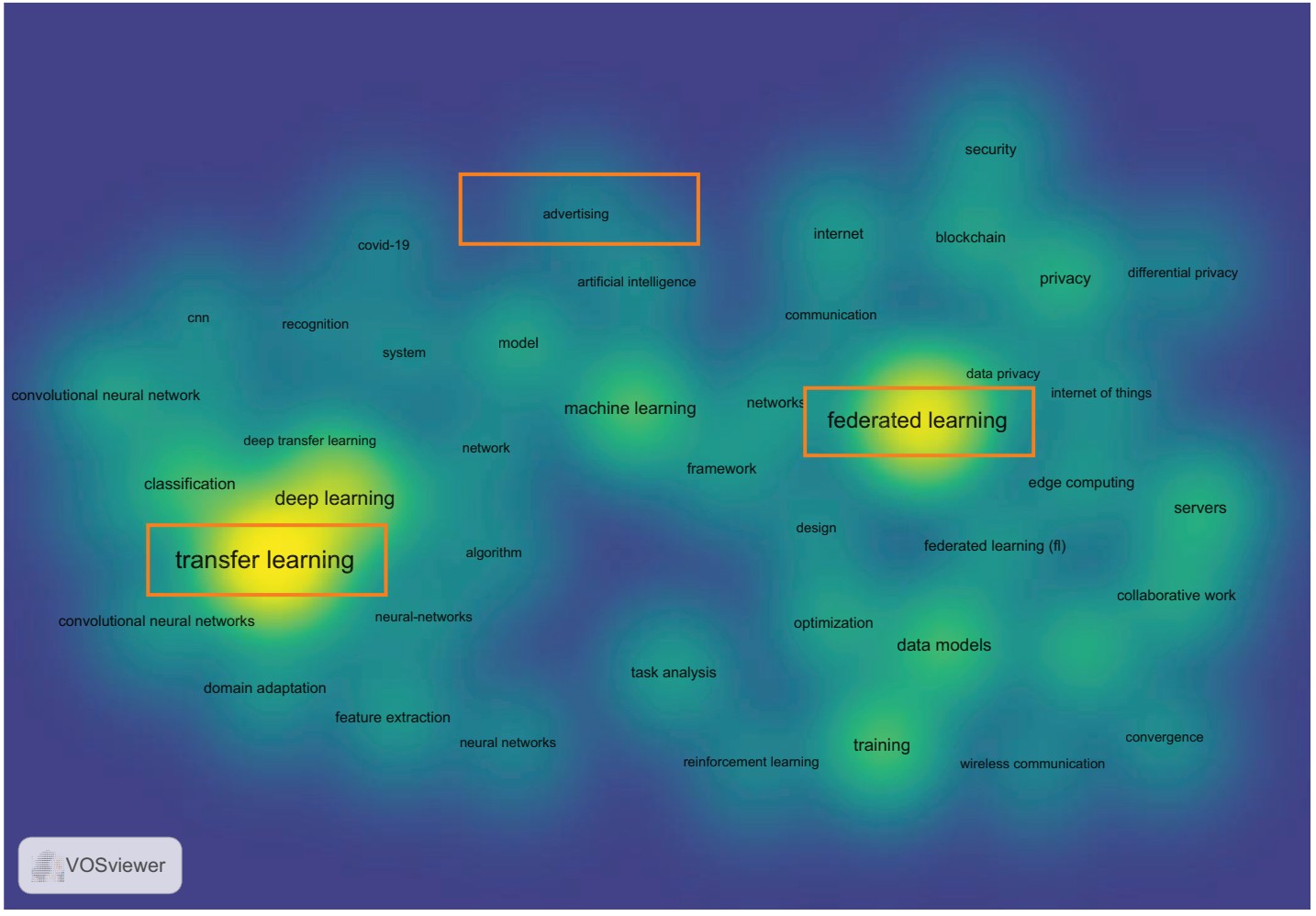

**Figure 1** The citation relationship between "Federated Learning", "Transfer Learning" and "Advertising".

$D_{public}$, the advertiser set A, and a specific advertiser $A_q$, where $q = \{1, 2, 3, \ldots\}$. The sample data for advertiser $A_q$ is represented as $A_q = X_1^{(q)}, X_2^{(q)}, \ldots, X_i^{(q)}$, and the label set for these data is represented as $Tag_q = Y_1^{(q)} \ldots, Y_i^{(q)}$. In each round of global iteration, the shared model of traditional FL cannot meet the needs of all advertiser nodes, and problems such as non-independent and identical distribution and heterogeneity exist, making it difficult for advertiser nodes to receive the benefits they deserve in the process of sharing information. Therefore, a personalized learning framework is needed to continuously update and improve FL. The personalized learning framework enables each advertiser to perform personalized training on their local ML model $D_{local}$ to improve personalized accuracy. At the same time, it ensures that different advertiser providers can be allocated the most reasonable returns when cooperating.
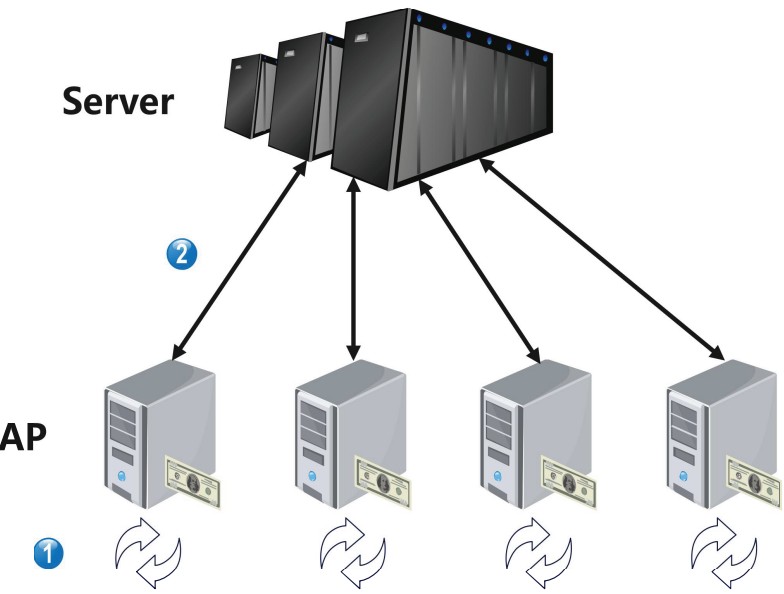

Server

AP

**Figure 2 System model.**

## Problem description

Assume that there are q advertisers participating in the calculation in the scenario of advertising targeting delivery, and each advertiser has its own local dataset $D_{local}$ at the beginning, which we denote as $\left\{X_i^{(1)} \in D_{local}^1\right\}, \left\{X_i^{(2)} \in D_{local}^2\right\},..., \left\{X_i^{(q)} \in D_{local}^q\right\}$, $i = \{1, 2, 3...q\}$. At the same time, there also exists a public dataset $D_{public}$ in this scenario. Since advertising targeting delivery data is mostly numerical, we choose long short-term memory (LSTM) as the core model. For the commonly used recursive network model (RNN), there are problems of gradient disappearance and gradient explosion when solving the problem of long-term correlation, and the role of the cell state is to continuously transfer specific information to the RNN. In this way, these problems can be effectively solved, and can be well applied to the scenario of advertising targeting delivery. LSTM is generally composed of forget gates, input gates, and output gates.

The role of the forget gate is to decide to retain or delete the information in the network output $h_{t-1}$ of the previous output in the time series.

$$R_t = \sigma\left(W_r^T \begin{bmatrix} h_{t-1} \\ x_t \end{bmatrix} + b_r\right) \tag{1}$$

In this, $W_m$ represents the weight of the forget gate; the superscript $T$ represents transpose; $b_m$ represents the bias of the forget gate; $h_{t-1}$ is the network input at time $t-1$; $x_t$ is the network input at time $t$; $\sigma(\cdot)$ represents the sigmoid activation function, which is used to ensure that the output values are between [0,1], and its expression is:

$$\sigma(x) = \frac{1}{1 + e^{-x}} \tag{2}$$

The role of the input gate is to calculate the weights for preserving and merging information into the state cell, which is to add new sequence information into the state cell. The expression for the input gate is as follows:

$$U_t = \sigma\left(W_u^T\begin{bmatrix} h_{t-1} \\ x_t \end{bmatrix} + b_u\right) \tag{3}$$

$$\widetilde{G_t} = tanh\left(W_g^T\begin{bmatrix} h_{t-1} \\ x_t \end{bmatrix} + b_g\right) \tag{4}$$

In which, $W_i$ represents the weight of the input gate; $W_c$ represents the weight in the tanh function; the superscript $T$ denotes transpose; $b_i$ represents the bias of the input gate; $b_c$ represents the bias in the tanh function. The expression of tanh(x) is:

$$\tanh(x) = \frac{e^x - e^{-x}}{e^x + e^{-x}} \tag{5}$$

The expression for the unit state $c_t$ at time $t$ is as follows:

$$g_t = f_t g_{t-1} + i_t \widetilde{G_t} \tag{6}$$

The function of the output gate is to determine how much of the information stored in the memory module is passed on to the next time step and the output module. In LSTM model, the output is calculated using the output gate and a tanh function. Therefore, the prediction output at time $t$, $h_t$, is calculated using the output gate and a tanh function.

$$O_t = \sigma\left(W_o^T\begin{bmatrix} h_{t-1} \\ x_t \end{bmatrix} + b_o\right) \tag{7}$$

$$h_t = O_t \tanh(g_t) \tag{8}$$

The function for the prediction accuracy of the personalized learning model can be inferred, where $W_o$ represents the weight of the output gate and $b_o$ represents the bias of the output gate.

$$Accuracy = \frac{1}{N}\sum_{i=1}^{N} \sigma\left(W_o^T\begin{bmatrix} h_{t-1} \\ x_t \end{bmatrix} + b_o\right)/Y_t \tag{9}$$

On this basis, we can calculate the user model loss function of the framework to be:

$$L(\omega) = L_i\left(\omega_i, D_{local}^i\right) + \lambda \cdot \sum_{D_{public}} L_{KL}\left(h_t^{(i)}, Y_i\right) \tag{10}$$

where $L_{KL}(\cdot, \cdot)$ is the Kullback–Leibler divergence. Additionally, by calculating the AUC value for the complete global model used for training, the dataset is set to a total of $M$ positive samples, $N$ negative samples, and $M + N$ predicted $\hat{Y}_{t,i}$ to evaluate the model's performance, $\lambda$ represents the transfer rate. $L_i(\cdot, \cdot)$ is the local loss.

$$AUC = \frac{\sum_{i=1}^{N} 1\{\hat{Y}_{t,i} = Y_{t,i}\}}{M} \tag{11}$$

## Problem summary

Through the introduction of the above three indicators, we can obtain the unified training goal of our personalized learning framework, through the transfer learning C-means clustering algorithm based on the maximum mean difference to accurately classify advertisers and the dynamic selection algorithm to eliminate asynchrony, so we can get the maximum accuracy and the minimum loss value, which is also expressed by the follows.

$$max\left\{Accuracy = \frac{1}{N}\sum_{i=1}^{N} \sigma\left(W_o^T\begin{bmatrix} h_{t-1} \\ x_t \end{bmatrix} + b_o\right)/Y_t\right\} \tag{12}$$

$$min\left\{L(\omega) = L_i\left(\omega_i, D_{local}^i\right) + \lambda \cdot \sum_{D_{public}} L_{KL}\left(h_t^{(i)}, Y_i\right)\right\} \tag{13}$$

$$AUC = lim\frac{\sum_{i=1}^{N} 1\{\hat{Y}_{t,i}\}}{M} \to 1 \tag{14}$$

$M$ is the number of positive samples, which means serving the right amount of ads.

Faced with the heterogeneity of various aspects among advertisers and the implementation of targeted advertising delivery, the proposed Fed-TLKD is expected to achieve higher accuracy, lower loss values, and better $AUC$ values. This will lead to a faster convergence between local models and global models and significantly improve the efficiency of personalized learning models.

The symbols used in this study are shown in Table 2.

## FED-TLKD FRAMEWORK

### Overall introduction of the framework

The Fed-TLKD framework is divided into two modules: Advertiser-C-means clustering with transfer learning based on maximum mean difference, and Central Server-federated learning algorithm based on knowledge distillation and weight correction.

- Our approach involves the utilization of transfer learning and clustering optimization algorithms, which are executed by an access point (AP). This approach is particularly relevant to ad targeting datasets, which may contain text data describing the ad content or evaluations of targeted users. Text data in this field have unique characteristics, and we have developed a feature-based transfer learning training algorithm to ensure that global ML parameters in different fields do not conflict and that scarce private data does not negatively impact experiments. Our approach emphasizes privacy protection while still achieving excellent results through optimized algorithms.

**Table 2 Symbols and explanation.**

| Symbols | Explanation | Symbols | Explanation |
|---|---|---|---|
| $D_{local}$ | Local dataset | $b_r$ | The bias of the forget gate |
| $D_{public}$ | Public dataset | $x_t$ | The network input at time $t$ |
| $A$ | Collection of ad providers | $\sigma(\cdot)$ | The sigmoid activation function |
| $q$ | Amount of advertising providers | $W_u$ | The weight of the input gate |
| $X_i$ | Sample | $W_g$ | The weight in the tanh function |
| $Tag$ | Data label | $b_u$ | The bias of the input gate |
| $Y_i$ | Sample label | $b_g$ | The bias in the tanh function |
| $i$ | Number of models | $g_t$ | The unit state at time $t$ |
| $h$ | Network output | $W_o$ | The weight of the output gate |
| $t$ | Iteration moment | $b_o$ | The bias of the output gate |
| $W_r$ | The weight of the forget gate | $L_{KL}(\cdot,\cdot)$ | The Kullback–Leibler divergence |
| $T$ | Transpose | $\hat{Y}_{t,i}$ | Predicted value |
| $M$ | The number of positive samples | $N$ | The number of negative samples |

- Our solution to the challenges of federated learning involves the implementation of a personalized federated learning algorithm. Traditionally, the effectiveness of federated learning relies heavily on the similarity of data distributions among clients and identical model architectures. However, in real-world scenarios, client data is often non-independent and non-identically distributed, which can result in global ML models trained through federated learning (FL) failing to meet the individual needs of each client or guarantee maximum benefit for all participating clients.

To overcome this challenge, our personalized federated learning algorithm leverages a two-pronged approach. Firstly, we compute an optimal aggregate model that takes into account the unique data distribution of each client. This ensures that the global model trained through FL is customized to the specific needs of each participant. Secondly, we update personalized model parameters based on the unique data of each client. This helps to further optimize the model for each individual participant and ensure maximum benefits are achieved.

## Advertiser provider-C-means clustering with transfer learning based on maximum mean difference

### Motivation for C-means clustering with transfer learning based on maximum mean difference

- Traditional clustering algorithms have been widely used in various engineering fields, such as fault detection and image segmentation. However, these algorithms often require high-quality and abundant samples to produce accurate results. In the case of advertiser providers, some advertisers may have a lack of samples or poor-quality samples, which greatly affects the performance of clustering algorithms and can cause

severe deviation in the central server's clustering, ultimately impacting the accuracy of global ML models.

- Clustering algorithms are traditionally improved based on five main perspectives: centroids, hierarchy, probability distribution, density, and graph. However, these clustering methods still require all data from the nodes and do not effectively address the problem of uneven sample data quality levels across nodes. Transfer learning can effectively solve this problem by automatically selecting useful information from sample data and improving the clustering performance of the global model by utilizing useful information in local model parameters. Additionally, by utilizing transfer learning based on ML parameters, it aligns well with the characteristics of federated learning, where only parameters are transmitted and not values.

- The transfer clustering algorithm based on parameters is highly affected by inter-domain differences. When the distribution difference between local ML parameters and global ML parameters is large, the effect of transfer learning may weaken or even result in negative transfer. Therefore, evaluating the distribution difference between local and global models is a crucial aspect of transfer clustering.

In order to improve its suitability for advertising targeting scenarios, this article introduces a novel algorithm, called CTLMMD, which is based on transfer fuzzy C-mean clustering and utilizes the maximum average difference criterion. CTLMMD draws inspiration from transfer methods that focus on feature representation and addresses the challenge of large inter-domain differences by projecting local and global model parameters data. By projecting these data into a common subspace, the difference between the distribution of local and global models is reduced, resulting in improved clustering robustness. Our experimental results demonstrate the effectiveness of our proposed method.

### Traditional transfer learning C-means clustering algorithm

In the traditional transfer fuzzy C-means clustering algorithm, while maintaining the basic structure of the objective function of the fuzzy C-means clustering algorithm, the clustering center learns from the local model parameters to obtain the clustering center between the local model parameters and the global model parameters. The correlation matrix is introduced into the objective function of the transfer fuzzy C-means clustering algorithm, and the objective function of the transfer fuzzy C-means clustering algorithm is obtained. This method can better classify the advertising data accurately, which is more conducive to the targeted delivery of advertisements.

$$minJ_{CTLMMD} = \sum_{i=1}^{N_t} \sum_{j=1}^{c_t} u_{ij}^{m_1} ||x_i - v_j||^2 + \lambda \sum_{i=1}^{c_\varepsilon} \sum_{j=1}^{c_t} r_{kj}^{m_2} ||\tilde{v}_k - v_j||^2 \tag{15}$$

In this equation, $x_i$ represents the $i$-th sample of the global model parameters, $v_j$ represents the $j$-th cluster center of the global model parameters, $\hat{v}_k$ represents the $k$-th cluster center of the local model parameters, $u_{ij}$ represents the membership degree of the

$i$-th sample to the $j$-th cluster center, $r_{kj}$ represents the correlation between the $k$-th cluster center of the local model parameters and the $j$-th cluster center of the global model parameters, $m_1$ and $m_2$ are fuzzy weighting coefficients, $\lambda$ represents the transfer rate, $C_z$ and $C_t$ respectively represent the number of cluster centers of the local model parameters and the global model parameters.

### C-means clustering with transfer learning based on maximum mean difference

The algorithm consists of five steps: (1) First, the global model parameters are projected into the common subspace; (2) then, the local model parameters' cluster centers are obtained through a prototype-based fuzzy C-means algorithm; (3) next, a projection matrix is learned so that the distribution differences between the projected local model parameters and the global model parameters are minimized as much as possible; (4) after that, in the common subspace, the projected source domain cluster centers guide the clustering of the projected target domain data; (5) finally, the clustering results of the global model parameters are obtained.

Consider there is a common subspace with a projection matrix $H \subseteq R^{(r*d)}$, (where $r$ is the dimension of the common subspace, determining the number of feature dimensions after data projection, and $d$ is the dimension of the original data). Assuming that the samples in the local model parameters and global model parameters have the same dimension $d$, the projection matrix $H$ can be used to project the local model parameters and global model parameters data into the common subspace. The $i$-th sample $Hx_{(i,z)}$ in the local model parameters and the $i$-th sample $Hx_{(i,t)}$ in the global model parameters, projected into the common subspace with feature dimension $r$, can be respectively represented as $Hx_{(i,z)}$ and $Hx_{(i,t)}$. Based on the maximum average difference criterion, the difference between the distribution of local model parameters and global model parameters in the common subspace can be calculated by the distance between the mean of the projected local model parameters samples and the mean of the projected global model parameters samples.

$$\text{Dist}(P_z, P_t) = \left\| \frac{1}{N_t} \sum_{i=1}^{N_t} Hx_{i,t} - \frac{1}{N_z} \sum_{i=1}^{N_s} Hx_{i,z} \right\|^2$$

$$= \frac{1}{N_t^2} \sum_{i=1}^{N_t} \sum_{j=1}^{N_t} Hx_{i,t} x_{j,t}^T H^T + \frac{1}{N_z^2} \sum_{i=1}^{N_t} \sum_{j=1}^{N_t} Hx_{i,z} x_{j,z}^T H^T \qquad (16)$$

$$- \frac{2}{N_t N_z} \sum_{i=1}^{N_t} \sum_{j=1}^{N_s} Hx_{i,t} x_{j,z}^T H^T$$

Formula (16) can be further simplified to

$$\text{Dist}(P_z, P_t) = H\Omega H^T, HH^T = I_{r*r} \qquad (17)$$

where $H$ is a unit matrix of dimension $r$. The constraint ensures that the projection matrix $H$ is an orthogonal matrix. By minimizing Formula (17), the domain difference between

the source domain and the target domain can be reduced, and the effect of transfer clustering can be improved.

By introducing Formula (17) into the objective function of the transfer fuzzy C-mean clustering algorithm, the objective function of the transfer fuzzy C-mean clustering algorithm based on the maximum average difference can be obtained as follows.

$$minJ_{CTLMMD} = \sum_{i=1}^{N_t}\sum_{j=1}^{C_t} u_{ij}^{m_1}||Hx_i - v_j||^2 + \lambda\sum_{i=1}^{C_z}\sum_{j=1}^{C_t} r_{kj}^{m_2}||H\tilde{v}_k - v_j||^2 + H\Omega H^T \tag{18}$$

wherein, $x_i$ represents the $i$-th sample in the target domain, $v_j$ represents the $j$-th cluster center in the target domain, $v_k$ is the $k$-th cluster center in the source domain, $\lambda$ is the transfer coefficient, used to control the degree of transfer learning. In order to handle the constraint of Formula (18), the Lagrange multiplier $\alpha_i$ and $\beta_k$ can be introduced to construct the Lagrange objective function of Formula (19):

$$J = J_{CTLMMD} + \sum_{i=1}^{N_t}\alpha_i\left(1 - \sum_{j=1}^{C_t} u_{ij}\right) + \sum_{k=1}^{C_z}\beta_k\left(1 - \sum_{j=1}^{C_z} r_{kj}\right) \tag{19}$$

The solution of Formula (19) is related to the matrices $U$, $H$, $V$ and $R$, so it is solved using an iterative optimization strategy. In the iterative algorithm, $U$, $H$, $V$ and $R$ are optimized one by one, that is, when one parameter is updated, the other parameters are fixed. $U$ represents a membership degree matrix, as it is indicated with $u_{ij}$, $H$ and $V$ are also expressed in the same way.

First, fix $U$, $H$, $R$ and take the partial derivative of $J$ with respect to $V$ and make the derivative equal to 0, we get:

$$v_j = \frac{\sum_{i=1}^{N_t} u_{ij}^{m_1} Hx_i + \lambda\sum_{k=1}^{C_s} r_{kj}^{m_2} H\tilde{v}_k}{\sum_{i=1}^{N_t} u_{ij}^{m_1} + \lambda\sum_{k=1}^{C_s} r_{kj}^{m_2}}, j = 1, 2, \cdots, C_t \tag{20}$$

Then, fix $V$, $H$, $R$, find the partial derivative of $J$ with respect to $U$, and make the partial derivative be 0, we get:

$$u_{ij} = \frac{(Hx_i - v_j)^{-2/(m_1-1)}}{\sum_{l=1}^{C_t}(Hx_i - v_l)^{-2/(m_1-1)}}, i = 1, 2, \cdots, N_t, j = 1, 2, \cdots, C_t \tag{21}$$

Then, fix $U$, $H$, $V$, find the partial derivative of $J$ with respect to $R$, and make the partial derivative be 0, we get:

$$r_{kj} = \frac{(H\tilde{v}_k - v_j)^{-2/(m_2-1)}}{\sum_{l=1}^{C_t}(H\tilde{v}_k - v_l)^{-2/(m_2-1)}} \quad k = 1, 2, \cdots, C_s, j = 1, 2, \cdots, C_t \tag{22}$$

The iteration of the projection matrix $H$ is more complicated, and some symbolic representations are introduced here first:

$$\overline{U}_1 = [u_{11}, \cdots, u_{i1}, \cdots, u_{N_t 1}] \in R^{1 \times N_t}$$
$$\overline{U} = [\overline{U}_1, \overline{U}_2, \cdots, \overline{U}_{C_t}] \in R^{1 \times C_t \cdot N_t}$$
$$\widehat{U} = \mathrm{diag}(\overline{U}) \in R^{C_t \cdot N_t \times C_t \cdot N_t}$$
$$\overline{R}_1 = [r_{11}, \cdots, r_{k1}, \cdots, r_{C_z 1}] \in R^{1 \times C_z} \tag{23}$$
$$\overline{R} = [\overline{R}_1, \overline{R}_2, \cdots, \overline{R}_{C_t}] \in R^{1 \times C_t \cdot C_z}$$
$$\widehat{R} = \mathrm{diag}(\overline{R}) \in R^{C_t \cdot C_z \times C_t \cdot C_z}$$

where

$$V_1 = \underbrace{[I_1, I_1, \cdots, I_1]}_{C_t} \in R^{N_t \times C_t \cdot N_t},$$
$$V_2 = \underbrace{[I_2, I_2, \cdots, I_2]}_{C_t} \in R^{C_z \times C_t \cdot C_z} \tag{24}$$

Among them, $I_1 \in R^{N_t \times N_t}, I_2 \in R^{C_z \times C_z}$, $I_1$ and $I_2$ are identity matrices.

$$Q_1 = \left[ q_{1,1}, q_{2,1}, \cdots, q_{1,C_t} \right] \in R^{r \times C_t \cdot N_t}$$
$$Q_2 = \left[ q_{2,1}, q_{2,2}, \cdots, q_{C_t} \right] \in R^{r \times C_t \cdot C_z} \tag{25}$$

where $q_{1,i} = \underbrace{[v_i, v_i, \cdots, v_i]}_{N_t} \in R^{r \times N_t}, q_{2,i} = \underbrace{[v_i, v_i, \cdots, v_i]}_{C_z} \in R^{r \times C_z}$. Substituting Formulas
(20)–(25) into Formula (18), the optimization problem about $H$ in Formula (18) can be transformed into

$$\min G(H) = \mathrm{tr}\left( (HX_t V_1 - Q_1)\widehat{U}(HX_t V_1 - Q_1)^{\mathrm{T}} \right)$$
$$+ \mathrm{tr}\left( \lambda(H\widetilde{V}_a V_2 - Q_2)\widehat{R}(H\widetilde{V}_s V_2 - Q_2)^{\mathrm{T}} \right) \tag{26}$$
$$+ \mathrm{tr}(H\Omega H^{\mathrm{T}})$$

Find the partial derivative of $G(H)$ with respect to $H$ as

$$\frac{\partial G}{\partial H} = 2\left( HX_t V_1 \widehat{U} V_1^{\mathrm{T}} X_t^{\mathrm{T}} - Q_1 \widehat{U} V_1^{\mathrm{T}} X_t^{\mathrm{T}} \right)$$
$$+ 2\lambda\left( H\widetilde{V}_s V_2 \widehat{R} V_2^{\mathrm{T}} \widetilde{V}_s^{\mathrm{T}} - Q_2 \widehat{R} V_2^{\mathrm{T}} \widetilde{V}_s^{\mathrm{T}} \right) + 2H\Omega \tag{27}$$

The gradient descent method can be used to calculate the optimal $H$. Set the initial value $H_0$ of $H$, and the iterative update process of $H$ is as follows.

$$H \leftarrow H - \eta \frac{\partial G}{\partial H} \tag{28}$$

## Central server-federated learning algorithm based on knowledge distillation and weight correction

### Motivation for federated learning algorithm based on knowledge distillation and weight correction

- Different advertising providers have model heterogeneity issue. As the rounds of iteration increase, the performance of the advertising providers will vary due to their own local ML parameters and the gap between them will also increase. Therefore, it is not a reasonable approach to perform equal updates for advertising providers with poor quality data samples.

- Knowledge distillation can transfer knowledge from one neural network to another by exchanging local ML model parameters instead of sample data, thus allowing for collaborative training of heterogeneous models while protecting privacy. Therefore, the heterogeneity issue of different advertising providers can be solved in this way.

- However, knowledge distillation does not fundamentally solve the problem of equal updates and does not consider the accuracy of the models of each advertiser provider. Therefore, we expect to use a non-linear weight correction function to ensure that the models of each advertiser node are updated non-equally during iteration.

Therefore, we propose a federated learning algorithm based on knowledge distillation and weight correction (KDWC). This algorithm effectively solves the heterogeneity of local ML parameters and the problem of equal updates among advertising providers. Similarly, this improvement will be proven in experiments.

### Traditional federated averaging algorithm

Due to the characteristics of federated learning, the central server does not receive any local data from the advertiser nodes, only local model parameters $w_q$ from the advertiser nodes. The central server aggregates the received local ML model parameters to obtain the overall parameter set $\{w = w_1, w_2, w_3, \cdots, w_q\}$, and updates the global model parameters. The traditional FedAvg model's weight update calculation formula for round (k + 1) is shown as follows.

$$w_{t+1} \leftarrow \sum_{i=1}^{I} \frac{q_i}{q} w_{t+1}^i \tag{29}$$

where $w_{(t+1)}$ is the updated global model parameters for the (k + 1)th round, $q_i$ is a certain advertiser node, $q$ is the total number of advertiser nodes, $w_{(t+1)}^i$ is the advertiser model parameters that are used for computation.

### Federated learning algorithm based on knowledge distillation and weight correction

Knowledge distillation is a model compression method based on the "teacher-student" idea, generally used to compress the knowledge of a complex model into a simpler model. First, we use a public dataset $D_{public}$ to train a teacher network, then use the output of the teacher network as the target for the student network to learn, thus using the local dataset

$D_{local}$ to train the student network to make its output increasingly close to the teacher network. Therefore, the loss of knowledge distillation includes two parts, one is the local loss, the other is the training loss. However, in the local loss, directly using the output of the softmax layer in the teacher network will filter out a lot of useful information, so we use a parameter $T$ $(T > 0)$ to smooth the probability distribution of each class, the softmax function is the special case of $T = 1$, the larger the value of $T$, the more the output probability distribution of the softmax layer will tend to be smooth, and the information carried by the negative label will be relatively enlarged.

$$L_{soft}(D_{local}) = L_{KL}\left\{\sigma\left(\frac{M_{stu}(D_{local})}{T}\right), \sigma\left(\frac{M_{tea}(D_{local})}{T}\right)\right\} \quad (30)$$

Among them, $M_{stu}$ represents the student model, $M_{tea}$ represents the teacher model, and the KL divergence is used to measure the gap between the two models.

The total loss of knowledge distillation can be expressed as:

$$L(D_{local}) = L_{cro}\{\sigma(M_{stu}(D_{local})), Tag\} + \lambda \cdot L_{soft} \quad (31)$$

Among them, $\lambda \in (0, 1)$ is a weighting coefficient, which is used to control the degree to which students learn from teachers, $L_{cro}$ is the cross-entropy loss function, and Tag is the label of the student-side data $D_{public}$.

Knowledge distillation migrates the training parameters of the teacher model to the local ML model for local training to complete the local model parameter update.

$$\bar{w}_i = w_i - \eta_1 \cdot \lambda \cdot \nabla_{w_i} L_{KL}(\hat{Y}_i^{t*}, Y_i, D_{public}) - \eta_2 \cdot \nabla_{w_i} L_i(w_i, D_{local}^i) \quad (32)$$

Among them, $w_i$ is the model parameter of advertiser $i$, $_1$ and $_2$ are the learning rate, $L_i$ $(w_i, D_{local}^i)$ is the loss function of advertiser $i$ on the local data $D_{local}^i$.

Then, through the weight correction function, the local model parameters are corrected and uploaded to the central server, thereby realizing the global model parameter update. The weight correction function looks like this:

$$w_i^* = \frac{1}{1 + e^{-\bar{w}_i}} + 1 \quad (33)$$

Thus, the calculation formula of the global model weight update in the (k + 1)th round is:

$$w_{t+1} \leftarrow \sum_{i=1}^{I} \frac{q_i}{q} w_{t+1}^i \quad (34)$$

### Pseudocode of the Fed-TLKD framework

This section presents the pseudocode for the Fed-TLKD framework, including the Advertiser-C-means clustering with transfer learning based on maximum mean difference

**Algorithm 1** Advertiser-C-means clustering with transfer learning based on maximum mean difference

**input:** $D_{public}, D_{local}, C_s, C_t, m_1, m_2, \lambda, \eta, n_{max}, \varepsilon$

**output:** $U$

1    FUNCTION ADVERTISER EXECUTES;

2    $\tilde{V}_k \leftarrow FCM(D_{public}, C_s)$

3    $U(0), R(0) \leftarrow Initialize(C_t)$

4    $H(0) \leftarrow Initialize(r)$

5    $t = 0$

6    **if** $t > n_{max}$ **then**

7    $$v_j = \frac{\sum_{i=1}^{N_t} u_{ij}^{m_1} H x_i + \lambda \sum_{k=1}^{C_s} r_{kj}^{m_2} H \tilde{v}_k}{\sum_{i=1}^{N_t} u_{ij}^{m_1} + \lambda \sum_{k=1}^{C_s} r_{kj}^{m_2}}, j = 1, 2, \cdots, C_t$$

8    $$u_{ij} = \frac{(H x_i - v_j)^{-2/(m_1-1)}}{\sum_{l=1}^{C_t} (H x_i - v_l)^{-2/(m_1-1)}}, i = 1, 2, \cdots, N_t, j = 1, 2, \cdots, C_t$$

9    $$r_{kj} = \frac{(H \tilde{v}_k - v_j)^{-2/(m_2-1)}}{\sum_{l=1}^{C_t} (H \tilde{v}_k - v_l)^{-2/(m_2-1)}} k = 1, 2, \cdots, C_s, j = 1, 2, \cdots, C_t$$

10    $H \leftarrow H - \eta \dfrac{\partial G}{\partial H}$

11    $t = t + 1$

12    return Acc, Loss, AUC to AP

13    END FUNCTION;

(refer to Algorithm 1) and Central Server-federated learning algorithm based on knowledge distillation and weight correction (refer to Algorithm 2).

Algorithm 1, the ad provider execution part mainly includes the initialization of parameters and clustering. Line 2 indicates the calculation of the clustering centers of the ad provider model parameters. Line 3–4 initializes the clustering-related matrices and projection matrices. Lines 6–10 calculate the clustering-related matrices and update the projection matrices.

Algorithm 2, the central server training module mainly includes the initialization of parameters and the training of the global ML model. Line 2 indicates the initialization of the server's global ML model parameters. Lines 6 and 7 indicates the sub-functions updated by the ad provider. Lines 9–11 revise the weights for the ad provider.

# EXPERIMENTAL RESULT

## Dataset description

We use two advertisement click-through rate prediction datasets, Ali-Display-Ad-Click (ADAC) and click-through rate data (CTRD), which are respectively sourced from the open data platform Alibaba Tianchi and the Kaggle online advertisement click-through rate prediction competition. The ADAC dataset includes 20 fields and 704,319 pieces of

| | **Algorithm 2** Central server-federated learning algorithm based on knowledge distillation and weight correction | |
|---|---|---|
| 1 | FUNCTION SERVER EXECUTES; | |
| 2 | Initialize $w_0$; | |
| 3 | **for** *each round $t = 1, 2, \ldots$* **do** | |
| 4 | $ms \leftarrow max(D \cdot V, 1)$ | |
| 5 | $SV \leftarrow (random\ set\ of\ ms\ advertisers)$ | |
| 6 | **for** *each advertiser $v \in SV$ in parallel* **do** | |
| 7 | $w_{r'}^v \leftarrow ClientUpdate(v, w_r)$ | |
| 8 | $U \leftarrow J_{CTMMD}(D_{public}, D_{local}, C_s, C_t)$ | |
| 9 | **for** *each advertiser $v \in U$* **do** | |
| 10 | $L(D_{local}^v) = L_{cro}\{\sigma(M_{stu}(D_{local}^v)), Tag\} + \lambda \cdot L_{soft}$ | |
| 11 | $\bar{w}_v = w_v - \eta_1 \cdot \lambda \cdot \nabla_{w_v} L_{KL}(\hat{Y}_v^{t*}, Y_v, D_{public}) - \eta_2 \cdot \nabla_{w_v} L_v(w_v, D_{local}^v)$ | |
| 12 | END FUNCTION; | |

data, with 80% of the dataset being used as the training dataset and the remaining 20% being used as the test dataset. *Gai et al. (2017)* used this dataset to propose an industrial strength solution with a model named Large Scale Piece-wise Linear Model (LS-PLM) for the main CTR prediction in an advertisement system. *Zhou et al. (2018)* proposed a deep interest network (DIN) for online advertisement and other industrial applications for advertisement click-through rate prediction. The CTRD dataset includes 17 fields and a total of 80,000 training data and 20,000 test data. These two datasets both predict the click-through rate of online advertisements and can well reflect the effectiveness of the advertisement being delivered to the designated audience, thereby verifying the ability of the Fed-TLKD model. Therefore, we choose to use these two datasets as the experimental datasets.

## Experimental environment and experimental parameter settings

### Experimental environment

The configuration of the experimental platform is as follows 12th Gen Intel(R) Core™ i7-12700H CPU@ 2.30 GHz, DDR4 2933 MHz 16 GB memory Windows 11 Home Edition 64-bit operating system, The experimental software uses Python 3.6, pytorch 1.4.0+cpu and Keras 2.4.3.

### Experimental parameter settings

This experiment uses LSTM to verify the advertising click rate dataset, with a LSTM framework structure consisting of input layer, hidden layer, and output layer. Specific experiment parameter settings are shown in Table 3.

**Table 3 Experiment parameter settings.**

| Parameter category | Parameter name | ADAC and CTRD |
|---|---|---|
| LSTM parameters | Number of data vector dimensions | 17 |
| | Hidden meta-dimensions | 128 |
| | Number of LSTM layers in series | 3 |
| | Linear layer 1 | 128 * 64 |
| | Linear layer 2 | 64 * 32 |
| | Linear layer 3 | 32 * 14 |
| | Batch first | True |
| | Dropout | 0.2 |
| | Bidirectional | True |
| Federated learning parameters | Global training rounds | 10 |
| | Local training rounds | 5 |
| | Initial advertisemets | 10 |
| | Data Distribution | Non-iid |
| | Learning rate | 0.01 |
| | Decay rate | 0.1 |
| | Processor | CPU |

## Experimental results

### The acc, loss, AUC results and computational time cost

Accuracy results

To better evaluate the performance of our framework, we compared the accuracy of the Fed-avg, Fed-sgd, Fed-prox, Fed-TLKD without CLTMMD (In response to: "Experimental Environment and Experimental Parameter Settings" C-means clustering with transfer learning based on maximum mean difference), Fed-TLKD without KDWC (in response to: "Experimental Results" federated learning algorithm based on knowledge distillation and weight correction), and Fed-TLKD. As shown in Fig. 3, the performance of the Fed-avg (*Chen, Orekondy & Fritz, 2020*), Fed-sgd (*Mudd et al., 2022*), and Fed-prox (*Kim & Hospedales 2023*) models remains poor with an increasing number of communication iterations of the global ML model. The Fed-TLKD without CLTMMD and Fed-TLKD without KDWC have good accuracy compared to the other three federated models, there is a lot significant improvement in the results. Only Fed-TLKD has constantly rising accuracy, and its precision is relatively high. Compared to other frameworks, the highest difference in model accuracy can reach 73%. The formula for maximum ACC is as follows:

$$max\left\{ Accuracy = \frac{1}{N}\sum_{i=1}^{N} \sigma\left( W_o^T \begin{bmatrix} h_{t-1} \\ x_t \end{bmatrix} + b_o \right) / Y_t \right\} \qquad (35)$$
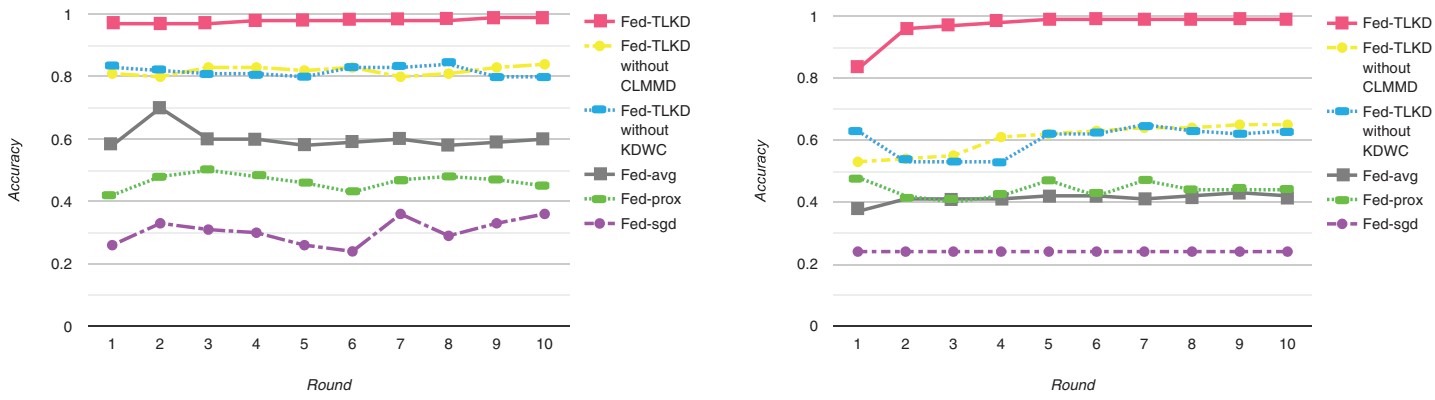

**Figure 3 Experimental model accuracy of ADAC dataset and CTRD dataset.**

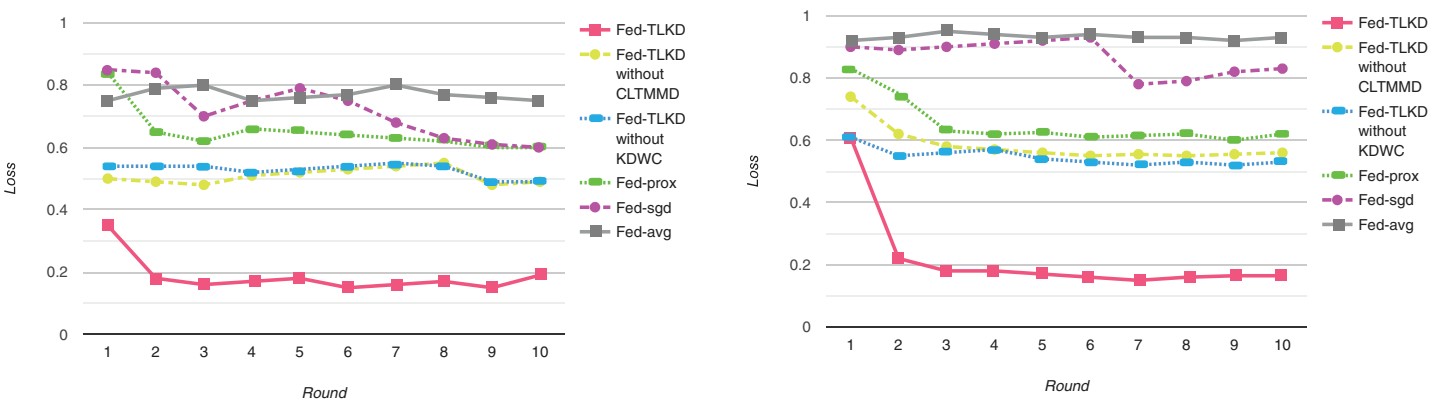

**Figure 4 Experimental model loss of ADAC dataset and CTRD dataset.**

## Loss Results

To better evaluate the performance of our models, we compared the loss values of Fed-avg, Fed-sgd, Fed-prox, Fed-TLKD without CLTMMD, Fed-TLKD without KDWC, and Fed-TLKD. As shown in Fig. 4, the performance of Fed-avg and Fed-sgd did not significantly improve with the increase in the number of communication iterations of the global ML model and they performed poorly. Fed-prox improved further, but there was still a significant difference. The Fed-TLKD without CLTMMD and Fed-TLKD without KDWC showed a consistent declining trend, but their performance still lagged behind that of the Fed-TLKD. The formula for minimum loss is as follows:

$$min\left\{ L(\omega) = L_i\left(\omega_i, D_{local}^i\right) + \lambda \cdot \sum_{D_{public}} L_{KL}\left(h_t^{(i)}, Y_i\right) \right\} \tag{36}$$

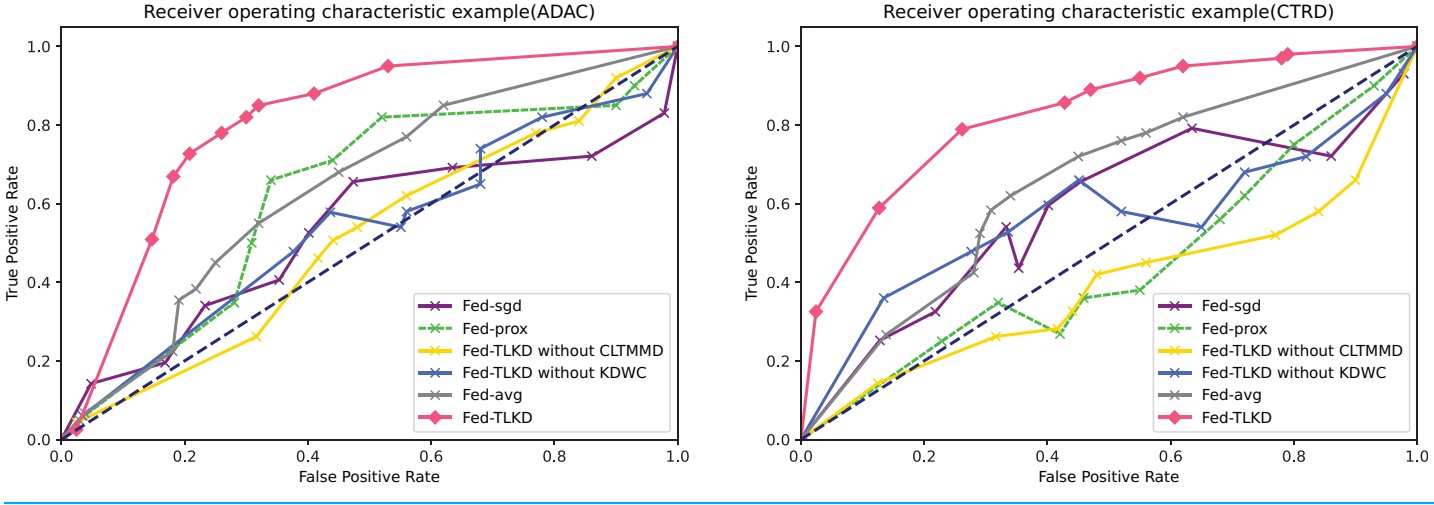

**Figure 5 Comparison of AUC.**

**Table 4 Time cost.**

| Training time for local ML models (unit: seconds) | ADAC | CTRD |
|---|---|---|
| Fed-avg | 4.9876 | 7.6481 |
| Fed-sgd | 4.9628 | 7.4572 |
| Fed-prox | 4.5263 | 7.2458 |
| Fed-TLKD without CLTMMD | 3.2658 | 6.1584 |
| Fed-TLKD without KDWC | 3.2494 | 6.2673 |
| Fed-TLKD | 3.2821 | 6.4862 |

AUC results

AUC accurately reflects the relationship between the true positive rate and false positive rate of a certain learner and represents the overall accuracy of detection. We also compared the above six frameworks. As shown in Fig. 5, it is clearly seen that the area under the curve of the Fed-TLKD is larger than the other five frameworks. The formula for AUC is as follows:

$$AUC = lim \frac{\sum_{i=1}^{N} 1\{\hat{Y}_{t,i}\}}{M \times N} \rightarrow 1 \qquad (37)$$

Computational time cost

The huge amount of data also affects the quality of frameworks, so being able to reasonably control the time cost also becomes an important index.

As shown in the results of Table 4, the communication time cost of the Fed-TLKD is almost the same compared to the other five, and the communication time cost of the remaining five models does not have much difference. Therefore, it can be proved that our Fed-TLKD framework is indeed better than the other five models.

## Conclusion

This article presents a cutting-edge federated learning framework, Fed-TLKD, which leverages transfer learning and knowledge distillation to tackle the challenges of data fragmentation and model variability in targeted advertising. Fed-TLKD features two algorithms: the C-mean clustering algorithm that incorporates maximum average difference transfer learning and a federated learning algorithm that employs knowledge distillation and weight adjustment. We evaluate the performance of Fed-TLKD on the ADAC and CTRD datasets for click-through rate prediction, and our results indicate that it outperforms other benchmark frameworks with a comparable computational cost. Consequently, Fed-TLKD represents a novel solution to the problems of data fragmentation, model variability, and data quality in targeted advertising and represents a significant advance in the field of federated learning.

### Funding

This work was supported by the Young and Middle-aged Teachers' Basic Ability Improvement of Guangxi Colleges in 2022 under Grant 2022KY1296. There was no additional external funding received for this study. The funders had no role in study design, data collection and analysis, decision to publish, or preparation of the manuscript.

### Grant Disclosures

The following grant information was disclosed by the authors:
The Young and Middle-aged Teachers' Basic Ability Improvement of Guangxi Colleges: 2022KY1296.

### Competing Interests

The authors declare that they have no competing interests.

### Author Contributions

- Caiyu Su conceived and designed the experiments, performed the computation work, prepared figures and/or tables, and approved the final draft.
- Jinri Wei performed the experiments, analyzed the data, prepared figures and/or tables, authored or reviewed drafts of the article, and approved the final draft.
- Yuan Lei conceived and designed the experiments, performed the experiments, analyzed the data, performed the computation work, prepared figures and/or tables, authored or reviewed drafts of the article, and approved the final draft.
- Jiahui Li conceived and designed the experiments, analyzed the data, performed the computation work, prepared figures and/or tables, authored or reviewed drafts of the article, and approved the final draft.

### Data Availability

  The raw data is available at Kaggle and Zenodo: https://tianchi.aliyun.com/dataset/56

Caiyu Su, Jinri Wei, Yuan Lei, & Jiahui Li. (2023). A federated learning framework based on transfer learning and knowledge distillation for targeted advertising-Ad Display/Click Data on Taobao.com dataset [Data set]. Zenodo. https://doi.org/10.5281/zenodo.8088629. https://www.kaggle.com/c/avazu-ctr-prediction

Caiyu Su, Jinri Wei, Yuan Lei, & Jiahui Li. (2023). A federated learning framework based on transfer learning and knowledge distillation for targeted advertising-Click-Through Rate Prediction Dataset (1.0) [Data set]. Zenodo. https://doi.org/10.5281/zenodo.8088573.

## Supplemental Information

Supplemental information for this article can be found online at http://dx.doi.org/10.7717/peerj-cs.1496#supplemental-information.

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
