# Peer review of "A federated learning framework based on transfer learning and knowledge distillation for targeted advertising"

_PeerJ Computer Science, doi:10.7717/peerj-cs.1496_

## Round 0.1 · original submission · Major Revisions

Dear Authors,

Good work! We have some suggestions to improve the article's quality. Please complete and re-submit for another review.

Reviewer 1 ·

Basic reporting

This paper proposes a C-means clustering algorithm based on maximum average difference to improve the evaluation of the difference in distribution between local and global parameters.

Experimental design

This proposed work has been tested on various datasets and its performance was evaluated using accuracy, loss, and AUC (Area under the ROC Curve) metrics.

Validity of the findings

Results showed that the framework outperformed other models in terms of higher accuracy, lower loss, and better AUC while requiring the same computation time.

Additional comments

1. Please improve the Abstract.
2. The Introduction section is very poor. In a research article, the introduction section must be very strong with the motivations of this paper, which is missing in this paper. Moreover, the disadvantages of the existing schemes must be discussed to motivate this new work.
3. Mentioned point-wise contributions are not convincing. The last paragraph of the Introduction should be the structure of the paper.
4. The Literature Review section is poor. The authors must include some more schemes. Also, the following papers must be cited to improve this section, as well as the Reference section:
a) An embedded vertical-federated feature selection algorithm based on particle swarm optimisation
b) Toward a self-supervised architecture for semen quality prediction using environmental and lifestyle factors
c) A boosting-aided adaptive cluster-based undersampling approach for treatment of class imbalance problem
d) ST-SIGMA: Spatio-temporal semantics and interaction graph aggregation for multi-agent perception and trajectory forecasting.
e) Experience replay-based deep reinforcement learning for dialogue management
f) Development of AR experiment on electric-thermal effect by open framework with simulation-based asset and user-defined input
g) Intelligent deception techniques against adversarial attack on the industrial system
5. In section 2, a table can be given to summarize the entire section.
6. What is the use of C-means clustering in the roposed scheme?
7. How the performance of the prediction is increased?
8. In Eq. (14), how the value of M is decided?
9. Search stage is completely unclear.
10. How Eq. (15) improves the performance of the model?
11. In section 5, add the “Experimental Environment” section.
12. What is the source of dataset? Whether it is authentic or not? Mention clearly.
13. How the results of Figure 3 are generated?
14. Technical details about results are missing.
15. How the training is done?
16. What is the novelty of this work? It is hard to identify from the current version of this paper.
17. Key terms of the equations must be defined.
18. Use a well-known software to draw the diagrams of the results section.
19. The organization of the paper must be improved. The paper must be formatted properly.
20. Improve the English language.
21. The Reference section must be improved significantly.

Reviewer 2 ·

Basic reporting

The biggest weakness of this paper is related to be poorly clear during some explanations in sections 2 and 3. Going section by section:

- Section 0 (Introduction and contribution) is OK. With this section, it is clear why this proposal is of high interest.

- Section 1 (Literature) requires (if it exists) adding information related to works focus on federated learning in which transfer learning is included, not necessarily in the advertising area. I mean, is there any work in which transfer learning is used in combination with federated learning?

- Section 2 and 3 have to be rewritten. Generally, there is a common problem: sometimes the amount of formulas and the abscence of extra explanations make the ideas difficult to understand. Formulas are not the problem, but it is required extra information to properly understand all stated during this sections. In fact, sometimes, there are some formulas in which some parameters are not stated, and it is more difficult to understand the formula. Two examples:

* In formula 10 it is not indicated what Li and lambda are (they are explained later in section 3).
* After formula 19 U, R and V are mentioned. We can consider then as the matrix generated with u_ij, v_j, etc. However, they are not defined. For instance, it could be stated that U represents a membership degree matrix, as it is indicated with u_ij, and after that consider the use of u_ij and U (same for V and R).

Furthermore, I am not sure if this AUC formula is right. I consider it is enough if authors indicate that AUC has to tend to 1. Going on with the needed of extra explanations, in section 3.2.3, figure X is referenced, but it does not exist. Probably, the inclusion of this and other figures/diagrams could make easier to properly understand all the paper.

Regarding the proposal, it is not clear how information obtained from clustering is used by the central Server-federated learning algorithm. In fact, inside algorithm 1 pseudo-code, it is said that U is the output (regarding membership degree) and in line 12 it is stated that the algorithm returns: Acc, Loss, AUC to AP. After that, U is used during algorithm 2 and it is mentioned that "for each advertiser in U". Are there advertisers in this matrix? After that, v (the advertiser) is not included in next formulations into this algorithm 2. Probably, for first for loop v is the letter to reference to each advertiser and in the second loop, i is the right letter, but it is not clear at all.

Finally, at this point, some minor corrections have to be made. Sometimes, abbreviations are used without indicating previously their meaning. For instance, AP or FL (FL is indicated in an image not in text). In pseudocode 1, in line 6 there is a mistake (¿).

Experimental design

In relation to experimental design, datasets used are properly indicated. There is a correction to make regarding the algorithms: references to Fed-avg, Fed-sgd and Fed-prox have to be included. I recommend the inclusion, if possible, of more datasets. In fact, if more datasets are added, statistical test could be made, which allows stronger conclusions.

Validity of the findings

Regarding the validity of the findings, the use of more datasets and statistical tests would make them stronger the conclusions. Attending to tables and graphichs, it seems clear that the proposal obtain better results in these two datasets than others. However, as indicated previously, the statistical test would make these conclusions stronger. In fact, authors indicate in line 398 that "there is no significant improvement in the results". However, this is said attending to visual results, not to statistical.

Additional comments

My recommendation is to make major revisions. The proposal is of high interest to solve some problems related to targeted advertising. However, into section 1, if there are ideas which combine both, federated and transfer learning, they should be included. Section 2 and 3 have to be rewritten to make them clearer (see recommendations made previously). Finally, more datasets and statistical tests should be included into section 4 (in order to extract conclusions from more than 2 datasets).

---

## Round 0.2 · Minor Revisions

Dear Author,

Good work!

Please complete the suggestions by the reviewers and re-submit.

Reviewer 1 ·

Basic reporting

This paper proposes a C-means clustering algorithm based on maximum average difference to improve the evaluation of the difference in distribution between local and global parameters.

Experimental design

This proposed work has been tested on various datasets and its performance was evaluated using accuracy, loss, and AUC (Area under the ROC Curve) metrics.

Validity of the findings

Results showed that the framework outperformed other models in terms of higher accuracy, lower loss, and better AUC while requiring the same computation time.

Additional comments

1. The Introduction section should be improved.
2. Mentioned point-wise contributions are still not convincing. They should be precise.
3. The authors must include some more recent references. The following references could be helpful to improve the Literature Review section, as well as the Reference section:
a) Deep generative inpainting with comparative sample augmentation
b) Multiview robust adversarial stickers for arbitrary objects in the physical world
c) Data accessing based on the popularity value for cloud computing
d) Novel multi-domain attention for abstractive summarisation
e) Research on face intelligent perception technology integrating deep learning under different illumination intensities
4. In the Literature Review section, a table can be given to summarize the entire section.
5. It is not clear how the performance of the prediction is increased?
6. Search stage is completely unclear.
7. Mention the use of Eq. (21).
8. How the results of Figure 4 are generated? Technical details about results are missing.
9. Key terms of the equations must be defined.
10. Use a well-known software to draw the diagrams of the results section.
21. Improve the English language.
12. The Reference section must be improved significantly.

Reviewer 2 ·

Basic reporting

Section 1 has been improved. The authors have included relevant research on the combination of transfer learning and federated learning.

Several corrections have been made to sections 2 and 3 to improve their comprehensibility.

Pseudo-codes regarding algorithms 1 and 2 have been improved.

Experimental design

The datasets and algorithms employed are adequately referenced now.

The references to other articles which used these datasets clarify the use of just these two datasets. However, as a suggestion, I go on with the idea that a higher number of datasets (for future works in this line) would allow more solid conclusions to be drawn.

Validity of the findings

The graphics employed have been improved. Once again, I suggest (for future works) using more datasets and statistical tests for more solid conclusions.

Additional comments

After major changes in the paper, I suggest accepting it. Thanks to the authors for responding to the suggestions made.

---

## Round 0.3 · accepted · Accept

Dear Authors,

Good work. You have addressed all of the required comments.